# Genome evolution and divergence in *cis*-regulatory architecture is associated with condition-responsive development in horned dung beetles

**Phillip L. Davidson** *, **Armin P. Moczek**

Department of Biology, Indiana University, Bloomington, Indiana, United States of America

* phidavid@iu.edu

**Data Availability Statement:** PacBio CCS data, Dovetail Omni-C data, and genome assemblies have been deposited on NCBI under BioProject accessions PRJNA966962, PRJNA966935,

## Abstract

Phenotypic plasticity is thought to be an important driver of diversification and adaptation to environmental variation, yet the genomic mechanisms mediating plastic trait development and evolution remain poorly understood. The Scarabaeinae, or true dung beetles, are a species-rich clade of insects recognized for their highly diversified nutrition-responsive development including that of cephalic horns—evolutionarily novel, secondary sexual weapons that exhibit remarkable intra- and interspecific variation. Here, we investigate the evolutionary basis for horns as well as other key dung beetle traits via comparative genomic and developmental assays. We begin by presenting chromosome-level genome assemblies of three dung beetle species in the tribe Onthophagini (> 2500 extant species) including *Onthophagus taurus*, *O. sagittarius*, and *Digitonthophagus gazella*. Comparing these assemblies to those of seven other species across the order Coleoptera identifies evolutionary changes in coding sequence associated with metabolic regulation of plasticity and metamorphosis. We then contrast chromatin accessibility in developing head horn tissues of high- and low-nutrition *O. taurus* males and females and identify distinct *cis*-regulatory architectures underlying nutrition- compared to sex-responsive development, including a large proportion of recently evolved regulatory elements sensitive to horn morph determination. Binding motifs of known and new candidate transcription factors are enriched in these nutrition-responsive open chromatin regions. Our work highlights the importance of chromatin state regulation in mediating the development and evolution of plastic traits, demonstrates gene networks are highly evolvable transducers of environmental and genetic signals, and provides new reference-quality genomes for three species that will bolster future developmental, ecological, and evolutionary studies of this insect group.

## Author summary

Phenotypic plasticity is the ability of a single genotype to produce multiple phenotypes in response to environmental variation and thus represents an important evolutionary

PRJNA966967 for Onthophagus taurus,
Onthophagus sagittarius, and Digitonthophagus
gazella, respectively. O. taurus ATAC-seq raw reads
have been deposited on NCBI under BioProject
accession PRJNA966925. Gene and repeat
annotations, comparative genomic results, and
ATAC-seq result files have been deposited on
Dryad under the DOI: 10.5061/dryad.wdbrv15t8.
Scripts utilized in this manuscript are available in
S1 Text and on Dryad.

**Funding:** This work is supported by National
Science Foundation grant no. 1901680 awarded to
APM and National Science Foundation
Postdoctoral Research Fellowship in Biology award
no. 2208912 to PLD. The funders had no role in
study design, data collection and analysis, decision
to publish, or preparation of the manuscript.

**Competing interests:** The authors have declared
that no competing interests exist.

mechanism for organismal adaptation and diversification. In this study, we investigate the
genomic basis for phenotypic plasticity in *Onthophagini* horned dung beetles, a highly
diverse (~2500 species) tribe of beetles renowned for their developmental plasticity. We
assemble and annotate chromosome-level genome assemblies for three *Onthophagini* spe-
cies and identify examples of gene family and coding sequence evolution potentially asso-
ciated with nutrition-responsiveness in this insect group. When then compare chromatin
conformation underlying head horn development and specification in *Onthophagus tau-
rus*, a species with dramatic nutrition- and sex-based differences in horn shape. We find
chromatin accessibility likely plays a critical role in specifying nutritional and sexual horn
dimorphisms in this species. Further, we show nutrition- and sex-responsive horn devel-
opment are controlled by largely distinct, rather than shared, regulatory architectures, and
the acquisition of lineage-specific regulatory elements may have played an outsized role in
the evolution of nutrition-responsive development of this trait. Our results highlight the
significance of chromatin accessibility and regulatory element activity in the regulation of
plastic phenotypes and the highly-evolvable nature of developmental gene networks.

## Introduction

Phenotypic plasticity is the capacity of a single genotype to produce multiple phenotypes in
response to environmental variation and constitutes a ubiquitous property of multicellular life
[1]. Plasticity is thought to be an important driver of adaptation, allowing organisms to main-
tain high fitness in the face of environmental variability, as well as of diversification via evolu-
tionary changes in the genetic architectures underlying plastic trait formation [2].

The ecological and evolutionary significance of phenotypic plasticity has received much
attention, and diverse genes and signal transduction pathways have been identified as impor-
tant mediators of plastic development across biological systems [3]. In addition to coding
sequence, epigenetic modifications such as histone marking are predicted to provide impor-
tant mechanisms of plastic gene expression regulation (for reviews, see refs [4–5]). Further,
recent quantitative trait locus (QTL) analysis combined with genome editing by CRISPR-Cas9
has begun to establish first causal connections between several *cis*-regulatory elements and the
plastic development of nematode feeding structures [6]. Yet despite these advances, the geno-
mic basis underlying developmental plasticity and its evolution, and in particular the role of
the non-coding genome and chromatin architecture in regulating conditional responses in
trait formation, remain largely unknown.

One group of animals that exhibit an extreme degree of phenotypic plasticity are the true
dung beetles (Scarabaeinae), a hyper-diverse clade (>6000 extant species) [7] found on every
continent except Antarctica. The extraordinary evolutionary success of this group is attribut-
able at least in part to their ability to exploit an abundant resource inaccessible to most other
insects–dung. For nearly every species, the acquisition and utilization of dung is essential to
each aspect of their life history. This includes not only consuming dung as a food source
(coprophagy), but also as a resource for larval food provisioning and nest construction, thereby
enabling a single offspring to complete development from egg to adult within the confines of
an underground brood ball. One key adaptation aiding in this strategy is a highly diversified
degree of nutrition-responsive (plastic) development. In the case of dung beetles, nutrition-
responsive development is a flexible developmental response to variable and limited larval
food quality and quantity, resulting in a wide range of adult body sizes, which in turn has
fueled the evolution of alternative, body size-dependent morphological, physiological, and

behavioral phenotypes [8]. Accordingly, phenotypic plasticity is predicted to be an evolutionary driver for many dung beetle adaptations. Furthermore, due to their diversity, abundance, pronounced environment-sensitive development, and unique feeding and reproductive traits, dung beetles have thus long served as important models for behavioral (e.g. status dependent selection and sperm competition models [9,10]), developmental (e.g. mechanisms of plasticity [11,12]), evolutionary (e.g. the origins of evolutionary novelties [13]), and ecological studies (e.g. meta-population theory [14], nutrient recycling, soil aeration [15,16]). However, despite the significance of dung beetles in both basic and applied science, a reference-quality genomic resource for any member of this insect group has so far been lacking and thus limited studies examining the genetic basis for dung beetle traits and adaptations.

Among the most conspicuous morphological trait of dung beetles are head horns—novel, highly diversified secondary sexual weapons used in reproductive competition [17]. Horns vary tremendously in shape, size, and number across and within species, mediate widespread sexual dimorphisms, and exhibit a high degree of nutrition-responsive development among conspecific males [18]. Most commonly, horn development is limited to, and often exaggerated, in males while females are hornless, though in some species both sexes may develop horns or–on rare occasions–horn development may be sex-reversed and more pronounced in females [19]. Here, nutrition responsiveness of horn formation provides an important axis of diversification, with horn size-body size scaling relationships ranging from isometric to positively allometric (exaggerated) to polyphenic. In polyphenic species, larval nutrition channels male development toward one of two alternate, and discretely different ontogenetic outcomes: fully horned *major* males (which as adults engage in aggressive combat to secure matings) or smaller-sized, nearly fully hornless *minor* males (which engage in sneaking tactics and sperm competition) [17,19]. Intriguingly, head horns lack homology to any other appendage or body part [20], and as such qualify as an evolutionary novelty even by the strictest of definitions [21], yet gains, losses, and modifications to horn structure are common among even closely related species [22]. Thus, beetle horns exhibit a high degree of evolutionary lability and represent a powerful natural system for understanding how complex traits originate and diversify [12,13,20,22].

In this study, we begin by assembling and annotating chromosome-level genome assemblies for three Onthophagine dung beetle species: *Onthophagus taurus*, *Onthophagus sagittarius*, and *Digitonthophagus gazella* (Fig 1A). The Onthophagini is a highly speciose (~ 2500 extant species) tribe of dung beetles found worldwide and includes one of the most species-rich genera on Earth, *Onthophagus* [23]. Through a series of whole genome alignments and comparative genomic assays including other published beetle genomes, we identify sequence gains and changes putatively associated with adaptations unique to dung beetle ecology and evolution.

We then utilize these three reference genomes to investigate the role of the non-coding genome in the development and evolution of sex- and nutrition-dependent horn formation across our three focal species. We chose these three species because collectively they embody a remarkable diversity in head horn development reflective of much of the diversity contained within the clade, including: a highly exaggerated sexual dimorphism and male polyphenism in *O. taurus*; a rare example of sex-reversed horn development in *O. sagittarius* females paralleled by a secondary loss of horn polyphenism in males; and a modest sexual dimorphism and male polyphenism in the more distantly related *D. gazella* thought to reflect ancestral character states (Fig 1B). Specifically, we apply genome-wide chromatin accessibility assays to investigate the *cis*-regulatory basis for sex- and nutrition-biased horn formation in *O. taurus* and provide some of the first evolutionary insights into the chromatin architecture underlying nutrition-dependent morph determination and developmental plasticity broadly. Our approach and results open new ways of understanding how gene networks may evolve to generate novel structures and regulate environment-responsive development.

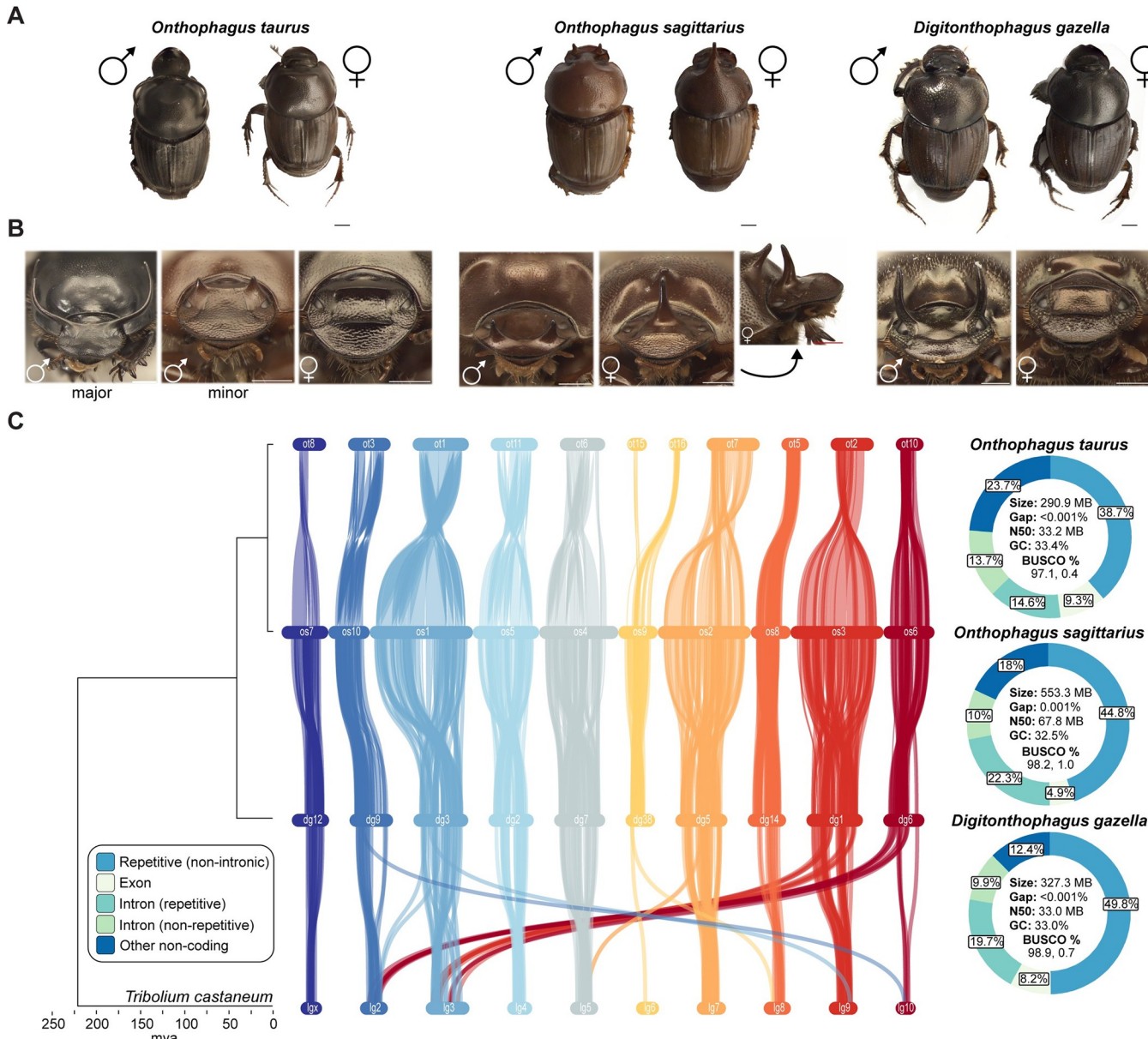

**Fig 1. Chromosome-level, reference genome assemblies for three dung beetle species.** (**A**) Images of adult male and female dung beetle individuals representing the focal species of this study: *Onthophagus taurus*, *Onthophagus sagittarius*, *Digitonthophagus gazella*. Scale bar: 10 mm. (**B**) Intra- and interspecific head horn morphologies of these dung beetle species, including a striking nutrition-dependent male polyphenism in *O. taurus* and sex-reversed posterior horn development in *O. sagittarius*. (**C**) Link plot illustrating synteny relationships between genes of chromosome-length scaffolds of these three dung beetle species and the more distantly related model beetle species, *Tribolium castaneum*. Genome composition and summary statistics for each dung beetle species are provided on the right.

## Results and discussion

### Assembly of reference dung beetle genome assemblies

Chromosome-level scaffolds were constructed for each beetle species using a two-step assembly strategy: PacBio HiFi long read contig assembly followed by HiC proximity ligation sequencing. Importantly, each species' assembly was generated and annotated using identical sequencing and bioinformatic pipelines (see *Methods and Materials* for details), minimizing

technical bias during cross-species comparisons. The *O. taurus* assembly is 290.0 Mb in length and is composed of 53.3% repetitive elements (Fig 1C). Interestingly, the closely related *O. sagittarius* genome assembly is nearly twice as large (553.3 Mb), which is almost entirely attributable to expansion of non-coding, primarily repetitive (69.1% of assembly), sequence. The *D. gazella* assembly, on the other hand, revealed only a modest increase in genome size (327.3 Mb) relative to *O. taurus*. High contiguity and BUSCO single copy scores (97.1–98.9%) alongside low BUSCO duplication scores (0.4–1.0%) suggest accurate reference genome assemblies were achieved for each species (Fig 1C). The *O. taurus*, *O. sagittarius*, and *D. gazella* genomes have an estimated 18,389, 19,766, and 18,889 number of gene models, respectively, which is in the range of other published beetle genome assemblies compared in this study (Fig 2A) (12,873–23,987). At a genome-wide scale, gene composition is also comparable in terms of orthogroup membership (a set of orthologous genes with shared ancestry across species, see ref. [24]), including both the proportion of species-specific orthogroups (Fig 2B) and number of genes per orthogroup (Fig 2C). Therefore, we next compared individual orthogroup size and nucleotide sequence across ten beetle species to detect putatively adaptive changes to dung beetle coding sequence.

## Coding sequence evolution associated with dung beetle-specific adaptations

In total, we identified 19,407 orthogroups across the three *Onthophagini* species reported here and seven other coleopterans for which chromosome-scale resolution genomes (or close to) where available (Fig 2A). These seven non-dung beetle species represent a diverse set of feeding ecologies (e.g. predation: *Photinus pyralis* (firefly); phytophagy: *Leptinotarsa decemlineata* (Colorado potato beetle); wood feeding: *Anoplophora glabripennis* (Asian long-horned beetle)) and a range of phylogenetic distances relative to the Scarabaeinae. From these orthogroups, we identified evolutionary changes in gene content in dung beetles relative to species in other beetle families in two ways: 1) examination of rapidly evolving gene families within multi-copy orthogroups (14,414 orthogroups) using CAFE [25] and 2) testing for evidence of episodic diversifying selection among single copy orthologs (2,948 orthogroups tested) with BUSTED [26]. While gene family size changes and non-synonymous mutations are not necessarily indicative of adaptation, these analyses provide predictions of candidate genes and pathways for further assessment. We highlight evolutionary changes at five phylogenetic positions (Fig 2A): each beetle species tip, the ancestral node of *O. taurus* and *O. sagittarius*, and the ancestral node of all three dung beetle species examined here (*Ot-Os-Dg* node).

For each dung beetle species, we identified ~200–300 rapidly evolving (mostly expanded) orthogroups as well 110 inferred gains and losses at the *Ot-Os-Dg* node (Fig 2D and S2 Data). Here, we call rapidly evolving orthogroups as those with significant gains or losses at a dung beetle position but *no* significant changes (in the same direction) at any non-dung beetle point in the phylogeny. While rapidly evolving gene families detected at the tips of the phylogeny may represent species-specific copy number changes or adaptations, we focused first on changes inferred at the *Ot-Os-Dg* node (orange) which we predict may most closely represent changes in gene family size associated with the *Onthophagine* ancestor and dung beetle biology. Among these 110 rapidly evolving orthogroups at the *Ot-Os-Dg* node are gene families involved in metabolism, metamorphosis regulation, and odorant receptor genes (Fig 2D inset). Specifically, among the largest gene family expansions are genes involved in juvenile hormone (JH) production (JH acid O-methyltransferase and JH esterase), in line with earlier work suggesting this hormone may be involved in regulating male dimorphisms in *O. taurus* [27] and the broadly described importance of this pathway in insect development and

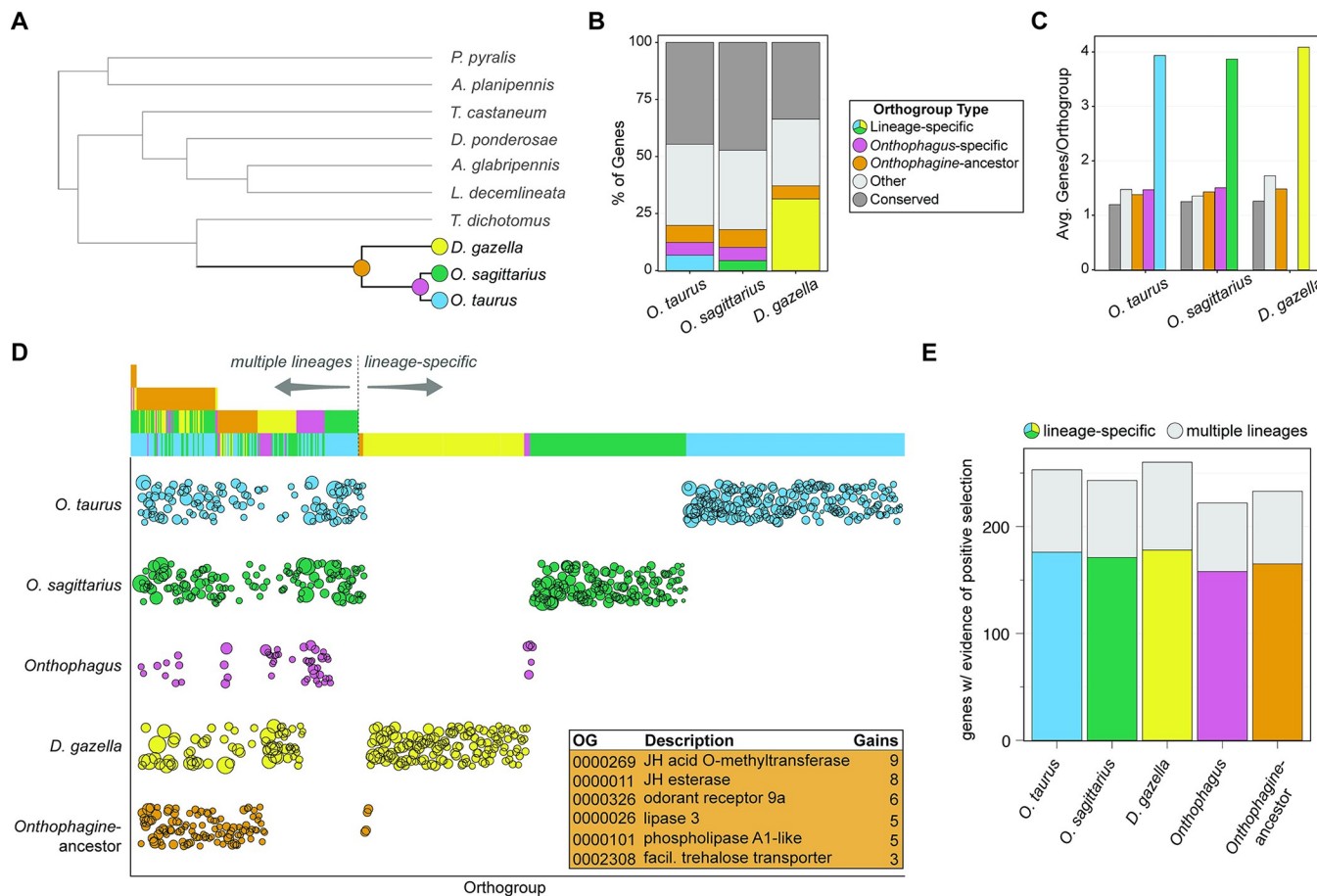

**Fig 2. Comparative genomics across the Coleoptera identify dung beetle-specific evolutionary changes in gene content. (A)** Phylogenetic relationships of ten beetle species included in comparative genomic analyses. We focus on five phylogenetic positions (colored dots) for identifying putative changes in gene content associated with dung beetle-specific traits: the species tips of 1) *O. taurus*, 2) *O. sagittarius*, and 3) *D. gazella*, 4) the ancestral node of *O. taurus* and *O. sagittarius*, and 5) the ancestral node of all three dung beetles species analyzed here. **(B)** Proportion of each dung beetles' gene set belonging to different orthogroup classifications. "Conserved" denotes orthogroups that include at least one gene from each of the ten beetle species included in the study. "Other" denotes orthogroups that have representation from some, but not all, of the ten beetle species. Onthophagine-ancestor, *Onthophagus*-specific, and lineage specific orthogroups are those with genes exclusively in 1) the three dung beetle species, 2) *O. taurus* and *O. sagittarius*, and 3) only one of the dung beetle species, respectively. **(C)** Average number of genes identified in each dung beetle species' genome per orthogroup type. **(D)** Rapidly evolving putative gene families (orthogroups) at each phylogenetic position of interest potentially associated with dung beetle adaptions. This includes gene families with significant expansions or contractions relative to the rest of the tree, which could occur in multiple lineages (stacked bars on left) or at a single phylogenetic position (single bars on right). Circle size denotes relative size of gene family and each bar on top corresponds to one circle within the plot. Orange inset lists orthogroups with some of the largest expansions at the estimated *Onthophagine* node, which we predict includes evolutionary changes to gene family size most representative of dung beetle-specific adaptations. **(E)** Number of genes with evidence of episodic diversifying (positive) selection at each phylogenetic position of interest but no evidence of selection at any non-dung beetle node. Colored bars denote genes with evidence of positive selection detected exclusively at that phylogenetic position, whereas gray bars denote genes with evidence of positive selection within multiple dung beetle lineages.

metamorphosis [28]. We also identified orthogroup expansions of genes involved in lipid (lipase 3; phospholipase A1-like) and carbohydrate (facilitated trehalose transporters) metabolism, which may be associated with diversification of nutrition-responsive development in dung beetles. Lastly, among the largest gene families within metazoans are those encoding odorant receptor (OR) proteins [29], the diversification of which is predicted to have aided in the evolution of insect terrestriality given their critical function in scent detection [30]. Expansion of the OR 9a family was inferred at the dung beetle ancestral node as well as separate expansions of OR 85d and OR 67c families in *O. taurus* and *O. sagittarius*, respectively, which may be associated with the localization and utilization of dung beetles' unique food resources.

However, like all gene family evolution analyses, associations of orthogroup expansions with dung beetle adaptations are speculative, and functional assessment of these genes and pathways will be necessary to validate the predictions mentioned above.

Next, contrasting orthogroup membership gains and losses between *O. taurus* and *O. sagittarius* offers a way of identifying rapidly evolving gene families potentially associated with the exaggeration (*O. taurus*) and secondary loss (*O. sagittarius*) of nutritional plasticity. We identified 42 orthogroups with significant gene number expansions in *O. taurus* and corresponding losses in *O. sagittarius* as well as 30 significantly expanded orthogroups in *O. sagittarius* that are reduced in *O. taurus* (S2 Data). Among the most dynamic of these orthogroups is one that includes targets of rapamycin (TOR), for which we observed 9 gene gains in *O. taurus* and 3 losses in *O. sagittarius*, paralleling the evolutionary exaggeration and secondary loss of male polyphenism observed in the two species, respectively. TOR proteins are a well characterized group of kinases that play a central role in eukaryotic cell growth and nutrition sensing as a part of critical developmental signaling pathways [31]. Given the correlative nature of this result and the fact that TOR copy number can vary widely among lineages, future comparative functional analyses of this pathway during dung beetle larval development will have to determine what role, if any, TOR protein signaling may play in nutrition-responsive development within and among species.

Lastly, we estimated evidence of gene-wide episodic diversifying (positive) selection in the coding sequence of 2,948 single copy orthogroups at the same five phylogenetic positions as above (Fig 2A) to identify changes in protein sequence possibly associated with dung beetle-specific adaptations. While we found 220–260 genes with evidence of positive selection on each position tested, we did not detect an enrichment of lineage-specific positive selection on any of the five lineages tested relative to other lineages (Fig 2E and S3 Data). However, within each of the dung beetle species reported here, we found evidence of positive selection in numerous developmental transcription factors including six homeobox proteins; for comparison, we detected evidence for positive selection in four homeobox proteins in the other seven non-dung beetle species *combined*. Taken together, these results document a remarkable number of non-synonymous substitutions in developmental regulatory genes of dung beetles, some of which may play adaptive roles in these species' development.

## Sex and nutrition-responsive regulatory elements underlie intraspecific diversity in beetle horns

Beetle head horns are novel structures, i.e. they lack obvious homology to other body parts, and are considered hotspots for evolutionary diversification due to the extraordinary morphological variation found both within and across species [18]. Previous work has identified critical transcription factors and developmental pathways that play a role in beetle horn development including *Doublesex* [11], Hedgehog signaling [32], and the insulin transduction pathway [33,34], among many others [35,36]. However, the role of the *cis*-regulatory elements (CREs) in mediating horn growth and diversification as well as transducing sex- and nutrition-responsive signals into alternative developmental phenotypes via gene network interactions remains poorly understood. To address this void, we measured chromatin accessibility using ATAC-seq (Assay for Transposase-Accessible Chromatin) in the epithelial cells that compose the dorsal, posterior head and thus the location in which large males develop prominent horns, small males develop horn rudiments, and females develop a conspicuous ridge (Fig 1B). We focused on individuals who had just completed the larval to pupal transition and replicated our approach in both males and females reared in high and low nutrition conditions to identify putative regulatory elements involved in sex- and morph-dependent horn morphogenesis.

It total, we identified 68,038 open chromatin regions (OCRs) in dorsal epithelial cells of *O. taurus* pupae, a portion of which we predict will contain CREs involved in developmental regulation of this body region. At a genome-wide scale, the chromatin landscape of high and low nutrition females is nearly indistinguishable and only 25 OCRs (0.037%) are significantly differentially accessible between nutritional conditions in this sex (S4 Data). This result may not be surprising given, like most dung beetle species, *O. taurus* females do not develop head horns or any other nutrition-responsive dorsal head phenotype. Accordingly, we grouped high and low nutrition female samples into a single female sample group (seven biological replicates) for downstream accessibility comparisons.

In stark contrast, we found 401 differentially accessible OCRs in primordial horn tissue of males reared in high vs. low nutritional conditions, including 250 OCRs more accessible in major males and 151 OCRs more accessible in minor males (Fig 3A: top). We categorize this set of OCRs as putative nutrition-responsive regulatory sites. Interestingly, over half (52.3%) of the 151 OCRs more accessible in minor males are located on the X chromosome (Scaffold 8), suggesting many loci responsible for regulating plastic horn growth are concentrated on this chromosome. Of note, one of the genes with the highest number of nearby nutrition-responsive OCRs is *doublesex* (Fig 3B: top; 4 OCRs), which previous work has demonstrated to be highly differentially expressed between male morphs [12] and to play an essential role in regulating both nutrition and sex-responsive horn development in *O. taurus* [11,37]. This concordance lends confidence to the accessibility data presented here and suggests regulation of *doublesex* expression, and by extension horn plasticity, may be associated with altered binding efficacy of *doublesex* regulatory factors to nearby CREs. More broadly, genes with nearby male nutrition-responsive OCRs more frequently exhibit differential mRNA expression between primordial horn tissue of male morphs than genes lacking nutrition-responsive OCRs (Fig 4D) (RNA-seq data from ref. [12]), though this association is not supported at a genome-wide scale (chi-square test: p = 0.15). This weak global correlation between OCR accessibility and transcriptomic divergence has been reported in other systems [38,39] and suggests alternative accessibility of OCRs is consequential for differential gene expression regulating plastic horn growth at some, but certainly not all, loci.

The number of male nutrition-responsive OCRs (401) is comparable to the number of differentially accessible OCRs between females and minor males (318) (Fig 3A: bottom). However, these differences are dwarfed by the 2,163 OCRs differentially accessible between females and major males (Fig 3A: middle; see Fig 3B: bottom for example OCRs nearby *foxp1*). One possible explanation for this observation is a compounding effect of two biological factors shaping the development of this body region: 1) sex and 2) exaggerated nutrition-responsive growth. Like in the comparison of male morphs, a large proportion of OCRs more accessible in females relative to major males are located on the X chromosome (55.7%), but surprisingly this sex chromosome enrichment is absent in the minor male and female comparison (Fig 3A). Thus, somewhat contradictory to expectations, genes and regulatory elements on the X chromosome may play an outsized role in male horn polyphenism development, whereas sex-based differences in the same trait may originate from loci more uniformly distributed across the genome.

## Distinct gene regulatory architectures underlie evolution of horn morphology

While horn development is dependent on both sex and nutritional signals, the degree of overlap between these gene networks underlying the evolution of intraspecific horn variation is mostly unknown. Earlier work suggested that sexually dimorphic horn formation may have

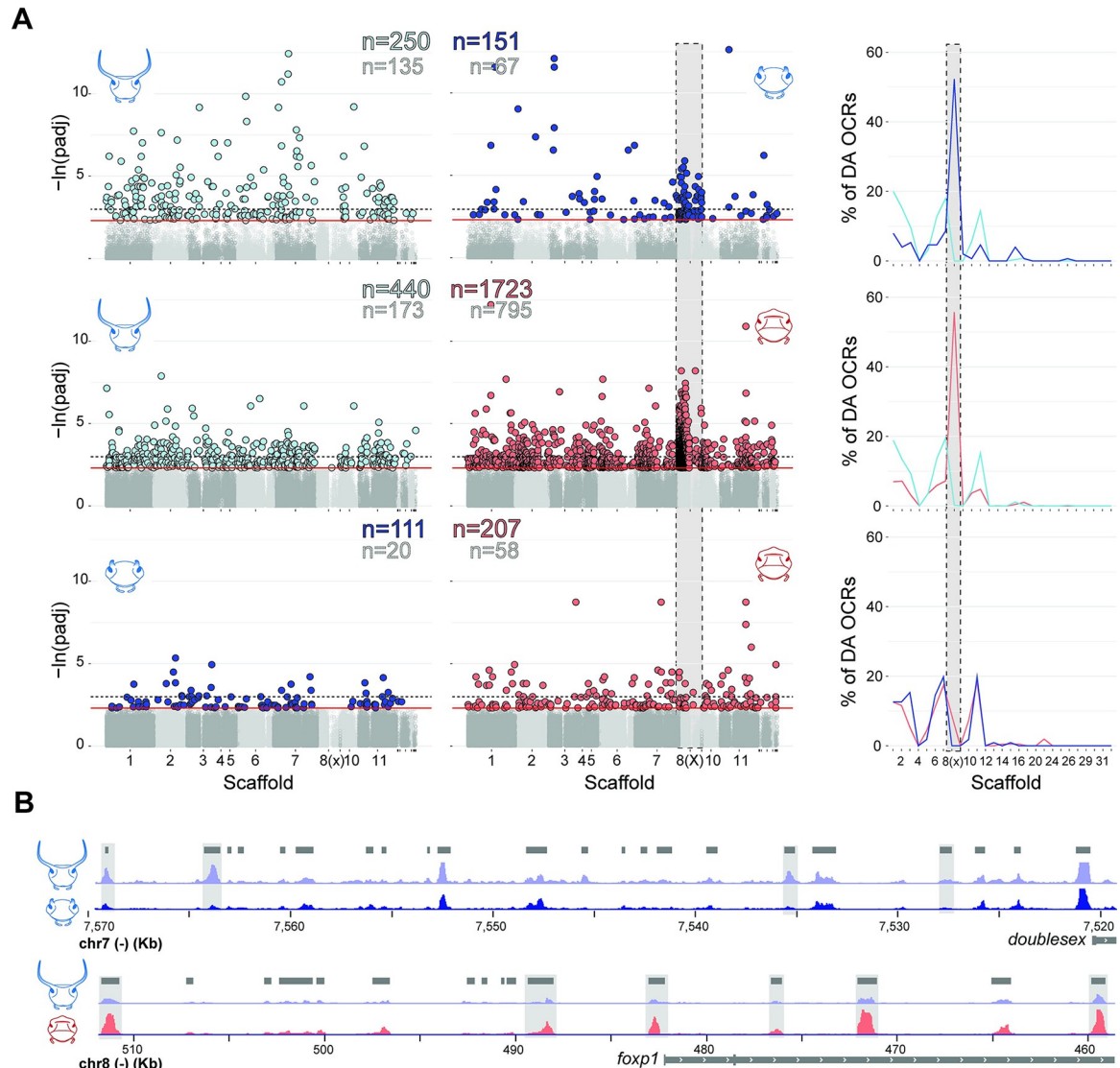

**Fig 3. Nutrition and sex-responsive open chromatin regions (OCRs) underlie intraspecific head horn variation in *O. taurus*. (A)** Manhattan plots of OCRs identified in primordial head horn epithelial cells, arranged by differential accessibility comparison: *top*- major (horned) males vs minor (hornless) males; *middle*- major (horned) males vs females; *bottom*- minor (hornless) males vs females. Gray points indicate non-differentially accessible OCRs for each comparison. Colored points indicate OCRs with significantly greater accessibility for each comparison. The red and dotted lines denote a false discovery rate (FDR) of 10% and 5%, respectively. The number of more accessible OCRs in each comparison are also provided at a 10% and 5% FDR (colored and gray numbers, respectively). Line plots on the right show the percentage of all differentially accessible OCRs that are located on each scaffold at the 10% significance threshold. Note the abundance of differentially accessible OCRs on the X chromosome (Scaffold 8) in minor male and female comparisons with major males. **(B)** Examples of genes with multiple male nutrition-responsive OCRs (top: *doublesex*) or sex-responsive OCRs (bottom: *foxp1*) upstream of their start codon (highlighted in gray) potentially involved differential gene expression via altered binding capacity of regulatory factors. Beetle head illustrations courtesy of Erica M. Nadolski.

evolved from sexually monomorphic horn development via the acquisition of mechanisms inhibiting horn development in females [18]. This perspective is supported by the observation that down-regulation of female *doublesex* isoforms induced horn development in otherwise hornless females [11]. Yet knockdown of *doublesex* expression mediates intersex phenotypes in diverse organisms and traits, and thus may not have directly driven the evolution of sexually dimorphic horn development in beetles. Earlier work also suggested that hornless-ness in both

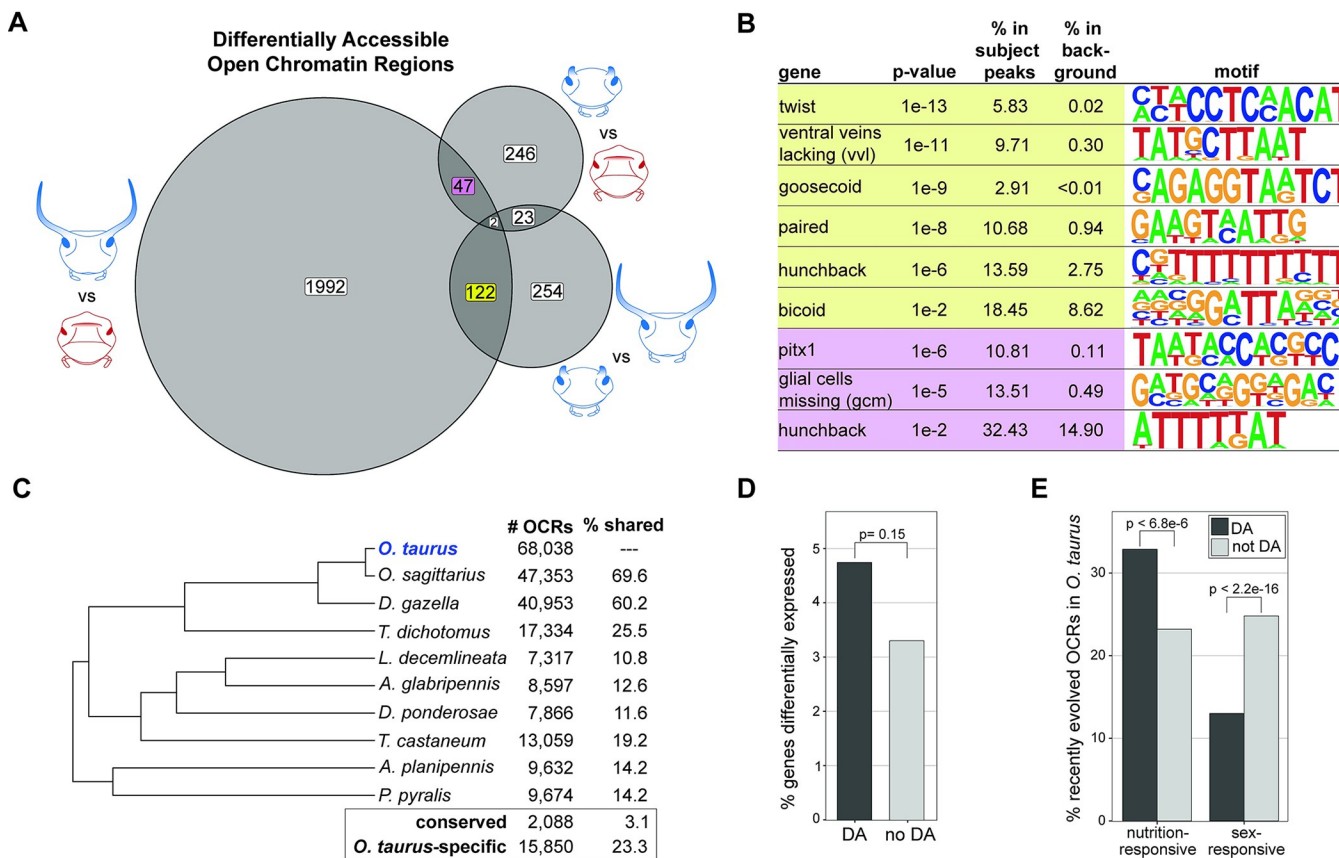

**Fig 4. Regulatory architectures underlying head horn shape are largely distinct.** (A) Venn diagram of shared and uniquely differentially accessible head horn OCRs identified in each pairwise comparison of *O. taurus* individuals. OCRs are detected in each tissue type. (B) Examples of binding motifs of developmental transcription factors significantly enriched in differentially accessible OCRs identified in both major (horned) male comparisons (yellow) or sex-based comparisons (purple). Shared major male OCRs (yellow) are predicted to contain regulatory elements especially important for exaggerated, nutrition-responsive horn development in this species. "% in background" refers to the percentage of OCRs containing a binding motif that are *not* differentially accessible in major males or between sexes. (C) Proportion of primordial head horn OCRs identified in *O. taurus* with orthology to loci in nine other beetle species included in this study. The "% shared" notes the percentage of the 68,038 OCRs identified on *O. taurus* that were also present in the genomes of one of the other beetle species. 3.1% of all OCRs were identified in *every* beetle species' genome (conserved set) whereas 23.3% of all OCRs were only detected in the *O. taurus* genome. (D) Proportion of genes that are differentially expressed between primordial head horn tissues of *O. taurus* male morphs that do or do not possess at least one nearby (nutrition-responsive) differentially accessible OCR. (E) Proportion of head horn OCRs found only in the *O. taurus* genome with differential accessibility according to male nutrition (left) or sex (right). Chi-square test results are presented in (D) and (E).

females and low nutrition males may be due to shared regulatory mechanisms [40], however, subsequent work has failed to support this notion. For example, manipulations of both insulin- and hedgehog signaling induce the formation of large horns in low nutrition males without affecting female dorsal head development [32,34].

In this study, we find that differentially accessible OCRs between developing horn tissue of male morphs and females are largely distinct from one another (Fig 4A), suggesting nutritional plasticity of *O. taurus* horn development is regulated by gene interactions largely separate from those governing sexual dimorphism. For example, of 318 OCRs differentially accessible between minor males and females, only 25 (7.86%) are also differentially accessible between major and minor males. Numerically, the largest overlap in differentially accessible OCRs is found when major males are compared to both females and minor males (124 OCRs), however, the same comparison identifies 254 differentially accessible OCRs (63.3%) as unique to the major vs minor male comparison and a remarkable 1,992 (92.1%) as unique to the major male vs female comparison (Fig 4A).

Importantly, both nutrition and sex-responsive OCRs are enriched for binding motifs of diverse developmental transcription factors relative to non-differentially accessible OCRs (Fig 4B), suggesting condition-dependent chromatin configuration provides a powerful molecular mechanism by which distinct suites of regulatory factors may exert their functions to determine head horn phenotypes. Similarly, "non-enriched" transcription factors that are known to play a role in specifying beetle head horn development also have putative binding sites within nutrition and sex-responsive OCRs (for a well-characterized example, see S5 Data for list of *Doublesex* hits). However, for these transcription factors, the proportion of binding sites residing within sex- and/or nutrition-responsive OCRs is *not* significantly different than the proportion of non-differentially accessible (background) OCRs with putative binding sites. This result may not be surprising given many transcription factors regulate gene expression in a variety of developmental and homeostatic contexts, which would require binding to distinct genomic loci. More generally, our results provide further support for the notion that sex- and morph-specific development are underlain by largely distinct regulatory landscapes and are thus developmentally and evolutionarily decoupled. Future work aimed at identifying which upstream regulatory molecules (e.g. pioneer factors) generate condition-dependent differences in chromatin accessibility will be critical to provide a more mechanistic understanding of the evolution of head horn regulation in this species.

## Recently evolved CREs play an outsized role in nutrition-responsive horn development

Most non-coding genomic elements such as *cis*-regulatory elements are generally subject to lower conservation rates relative to coding sequence across evolutionary time [41,42]. Therefore, we sought to understand what role, if any, recently evolved *cis*-regulatory elements play in *O. taurus* head horn development. Here, we are defining recently evolved (putative) regulatory elements as genomic regions (identified as larger OCRs from the ATAC-seq data) found in *O. taurus* but not detected in any other of the nine beetle genomes surveyed, which total 15,850 (23.3%) of all OCRs identified in this tissue type (Fig 4C) (see *Methods and Materials* for details on how orthologous regions were identified). By comparison, only 3.1% of all OCRs have an orthologous genomic region in each of the other 9 beetle species. We use the term "recently evolved" in lieu of "novel" when describing these genomic elements as we presently cannot distinguish between 1) truly novel regulatory sequences present only in *O. taurus* and 2) highly variable regulatory sequences for which synteny can no longer be assigned. Still, regardless of how these genomic elements originated, the relatively short evolutionary time examined (~5 my diverged from *O. sagittarius*) suggests these putative regulatory regions may be especially pertinent to horn development and evolution in this species.

When comparing chromatin accessibility between male morphs, we found 23.2% of *non*-differentially accessible OCRs have recently evolved in *O. taurus* (Fig 4E), which is nearly identical to the proportion of recently evolved elements among all 68,038 OCRs (23.3%, see above). Strikingly, the proportion of recently evolved regulatory elements increases to 32.9% among male nutrition-responsive OCRs in primordial horn epithelial tissue (Fig 4E). This enrichment of recently evolved regulatory elements lies in stark contrast to sex-responsive OCRs, for which only 13.0% of OCRs appear exclusively in the *O. taurus* genome (Fig 4E). Intriguingly, these recently evolved OCRs are highly enriched for repetitive elements like TEs (77.5% overlap at least one repetitive element) compared to conserved OCRs (26.7%), suggesting TE activity may have played a role in the origination of these putative regulatory regions (S1 Fig).

One explanation for this opposing relationship between nutrition- and sex-responsive OCRs may be attributable to the evolutionary history of the molecular mechanisms underlying

these two sets of phenotypes. Sexually dimorphic horn development is common not just in the *Onthophagini* but the family Scarabaeidae and thus likely originated early in scarab evolution. If so, we would not predict *recently evolved* regulatory elements to play an especially important role in the sex-biased development of this trait given the core of its developmental regulatory architecture evolved deep in this lineage's history. In contrast, nutritional plasticity, and in particular the extreme polyphenic development of *O. taurus* head horns, constitutes a more recently derived trait, whose diversification is more likely to be impacted by the acquisition of lineage-specific regulatory elements and the establishment of novel regulatory interactions these afford–a phenomenon that may be important for the evolution of novel traits in other systems as well.

## Conclusions

Nutrition-responsive phenotypic plasticity is a ubiquitous property of living systems and important biological mechanism for adaptation and trait diversification [1,2]. In this study, we first used comparative genomics to identify coding and non-coding changes in dung beetles associated with metabolite and developmental regulation that are potentially underpinning the evolution of nutritionally plastic phenotypes in these species. These biological pathways are involved in lipid and carbohydrate metabolism, juvenile hormone production, and embryonic transcriptional regulation, which collectively are now motivating new hypotheses for how nutrition-responsive development may have evolved in this group of insects.

We then carried out an in-depth genomic and developmental analysis of one of these traits, head horns–secondary sexual weapons with an extreme degree of inter- and intraspecific variation. Chromatin configuration assays suggest alternative forms of intraspecific horn variation (sexual dimorphism vs. nutritional plasticity) are instructed by discrete regulatory architectures, a conclusion corroborated by two independent pieces of support: 1) largely distinct sex- and nutrition-responsive head horn OCRs; and 2) a disproportionate amount of OCRs that are nutrition-, but not sex-, responsive and only found in the *O. taurus* genome. In other words, accessibility of putative *cis*-regulatory elements covaries extensively with sex and, in males, nutritional state, of which the latter set is disproportionally composed of recently evolved regulatory elements. Future work comparing the overlap of sex- and nutrition-dependent changes in regulatory element activity of other body regions with varying degrees of sex and nutrition responsiveness will be critical to contextualizing these results and understanding how gene regulatory mechanisms evolve across disparate traits.

One possible explanation for these findings may be linked to the idea that head horns lack obvious homology to other body parts in insects or non-insect hexapods. In contrast to legs, antennae, and mouthparts (which are all serial homologs of ventral appendages) and thoracic horns (which have recently been identified as wing serial homologs [13,20]), head horns appear to be unique elaborations of the dorsal head, and sufficiently individuated to have undergone remarkable diversification in shape, relative size, number, and in part, precise location. Lacking homology to other body parts may have promoted, permitted, or alternatively limited head horn development to the inclusion of novel regulatory interactions into gene networks due to the lack of pre-existing mechanisms that could be co-opted, and/or to minimize developmental constraint arising from pleiotropy. Comparing our findings on head horns to that of other diversified structures with established homology (such as ventral appendages or thoracic beetle horns) alongside other novel structures (such as butterfly wing spots or firefly lanterns) will be informative for understanding if the patterns of *cis*-regulatory evolution identified here are symptomatic for the early stages of morphological innovation in evolution.

## Methods and materials

### Genome sequencing and assembly

*O. taurus* individuals were collected from Chapel Hill, North Carolina, USA (35.936, -79.128), *O. sagittarius* were collected nearby Kilcoy, Queensland, AU (-26.951, 152.605), and *D. gazella* individuals were collected nearby Legonyane, South Africa. For each species, F0 individuals from these wild populations were crossed and offspring reared in artificial brood balls to the late pupal stage. Once this developmental stage was reached, a single male individual from each species were flash frozen in liquid nitrogen, stored at -80° C, and shipped overnight to Dovetail Genomics (Scotts Valley, CA, USA) on dry ice for library preparation and DNA sequencing.

A dual sequencing approach was implemented for *de-novo* genome assembly of each dung beetle species: 1) PacBio HiFi CCS long read sequencing for contig assembly and 2) Dovetail Omni-C proximity ligation sequencing (DNASe I digestion) for building contigs into chromosome-length scaffolds. PacBio coverage and read N50 lengths of (86.6X, 13.8 Kb), (30.8X, 12.2Kb), and (72.4X, 12.8 Kb) were achieved for *O. taurus*, *O. sagittarius*, and *D. gazella*, respectively, on a Sequel II platform. Omni-C libraries were sequenced on an Illumina HiSeqX platform to 30.8 million, 102.2 million, and 31.6 million 150 bp paired-end reads for each species, respectively. Contigs were constructed using the PacBio HiFi reads via *hifiasm v. 0.13* [43] under default parameters. Omni-C read pairs were aligned to the contig assembly using *bwa* [44] and scaffolds were assembled using the HiRise software pipeline [45]. Genome summary statistics are provided in Fig 1C. BUSCO v5.4.2 was implemented to estimate the completeness and redundancy of gene content within each assembly by referencing the "insecta_odb10" lineage dataset.

### Gene modelling and annotation

After the final version of each genome assembly was completed, we used RepeatModeler v2.0.1 [46] to create a library of repetitive elements for each species' genome. This library was then used to mask repetitive elements within each genome assembly with RepeatMasker v.4.1.1 using the most sensitive pre-set parameters (-s), which maximizes detection of repetitive elements at the expense of computational time. Soft-masked genome assemblies were used as input for proceeding gene modeling and annotation analyses.

We implemented the BRAKER2 [47] pipeline to generate an *ab-initio*, preliminary set of gene and isoform models for each dung beetle species, aided by the inclusion of RNA-seq data from each species[12] and the UniRef90 Ecdysozoan protein database (accessed Feb. 2022) [48]. The "-etpmode" pipeline includes a large suite of software including Augustus [49], Genmark-EP+ [50], Genmark-ET [51], DIAMOND [52], and SAMtools [53]. Gene models were assigned putative identification by BLAST-ing each model to the UniRef90 Ecdysozoan protein database and the *Onthophagus taurus* v. 2.0 protein models (e-value 1e-5).

Genome-wide synteny analysis was carried out using the gene-anchored MCScanX method [54] under default parameters. Pairwise alignments were performed in order of increasing phylogenetic distance starting with *O. taurus*, illustrated in Fig 1C: *Ot-Os*, *Os-Dg*, *Dg-Tc* following an all-to-all BLASTP similarity search carried out between each pair of species' protein models (-max_target_seqs 5, -evalue 1e-10). Synteny results from these analyses were then visualized using SynVisio [55].

### Comparative genomic analyses across ten Coleopteran species

Comparative genome analyses included the three newly assembled dung beetle genomes presented in this study alongside seven other high-quality beetle genomes representing a diverse

set of families across the Coleoptera. These genomes include the common eastern firefly *Photinus pyralis* [56], emerald ash borer *Agrilus planipennis*, red flour beetle *Tribolium castaneum* [57], mountain pine beetle *Dendroctonus ponderosae* [58], Asian longhorned beetle *Anoplophora glabripennis* [59], Colorado potato beetle *Leptinotarsa decemlineata* [60], and the Japanese rhinoceros beetle *Trypoxylus dichotomus* [61]. For all analyses of gene content between species, only the longest isoform of each species' gene was used. Protein models of these longest isoforms were input into OrthoFinder v.2.5.4 [24] to identify orthogroups across all ten beetle species, which served as the basis for analysis of rapidly evolving gene families and evidence of positive selection within coding sequence described below. Putative annotations for each orthogroup were assigned by BLAST-ing protein models (e-value < 1e-5) from each orthogroup to the NCBI non-redundant invertebrate peptide database (accessed November 2022) and filtering for significant hits to non-repeating protein descriptions (S1 Data).

## Gene family size evolution analyses

Nucleotide alignments of single copy orthogroups identified above were input into IQ-Tree v.2.2.0 [62] to generate an initial beetle species tree using the best fit model "JTT+F+R8" (1000 bootstrap replicates) identified by ModelFinder [63], and a time-calibrated species tree was estimated with r8s [64]. While only 10 taxa were included in this analysis and species tree determination was not a primary objective of this study, each node within this tree had 100% bootstrap support. Furthermore, its topology and estimated branch lengths are congruent with previously published, more in-depth phylogenomic analyses of Coleopteran relationships [65], providing confidence for this tree's inclusion into gene family evolution analyses. After filtering out exceedingly large orthogroups (5), 14,414 multi-copy orthogroups remained for rapid gene family size evolution analysis. We input these orthogroups and the species tree described above into CAFE v5 [25], while modelling for errors in gene family sizes associated with different assembly and annotation methods across the genomes included here [66]. Rapidly evolving gene families putatively associated with dung beetle specific traits (Fig 2D) are those orthogroups with significant expansions or contractions (p-value < 0.05) at a dung beetle species tip or ancestral node but *no* significant changes at any other non-dung beetle position (see S2 Data).

## Tests for episodic diversifying selection

To test for evidence for episodic diversifying selection within beetle coding sequence, nucleotide sequences of 2,660 single copy orthogroups were aligned across species using MUSCLE v5.1 [67]. Nucleotide alignments were input into BUSTED v.4.0 [26] while accounting for synonymous rate variation in the model to test for gene-wide evidence of positive selection for every orthogroup at six phylogenetic (foreground) positions: 1–3) each dung beetle species tip, 4) ancestral node of *O. taurus* and *O. sagittarius*, 5) ancestral node of all three dung beetle species reported here, and 6) *any* non-dung beetle position. P-values were adjusted using the Bonferroni method for multiple comparisons correction in R. Genes with evidence of positive selection putatively associated with dung beetle adaptation were those with significant evidence of selection (adjusted p-value < 0.05) at a dung beetle position and *no* evidence of positive selection coding sequence of any non-dung beetle species (S3 Data).

## ATAC-seq Sample and Data Preparation

*Onthophagus taurus* individuals were collected from Chapel Hill, North Carolina (same population as the individual used for genome sequencing) and kept in laboratory conditions as described in Moczek and Nijhout, 2002 at 24°C. Six female and three male adult individuals

were bred over the course of one week, after which brood balls collected and 1[st] instar larvae moved to artificial brood balls made in 12-well plates (an established protocol, detailed in ref. [68]). To simulate high and low nutritional conditions, larvae were fed manure from either grass-fed cows (high nutrition) or hay-fed cows (low nutrition) over the course of larval development (method detailed in ref. [69]). Plates were checked every 12 hours for pupating individuals. Distinguishing high and low nutrition males of *O. taurus* pupae is straightforward, as the fully developed horns of large (major) and horn rudiments of small (minor) male morphs become externally visible and easily recognized upon pupation. To distinguish high and low nutrition females, only pupae of the lower and upper quartile of the normal pupal weight distribution were selected for ATAC-seq sampling, i.e. pupae > 140 mg for the high-nutrition set and < 110 mg for the low nutrition set. Male pupae of corresponding weight classes invariably metamorphose into alternate horned and hornless morphs, respectively [70]. In total, we collected 16 biological replicates for ATAC-sequencing: 7 females (including 4 high and 3 low nutrition), 4 high nutrition males, and 5 low nutrition males. One high nutrition male and one low nutrition female ATAC-seq libraries required greater than 20 PCR cycles to properly amplify (see details in next paragraph) and as a result, were not sequenced due to poor library construction and complexity.

Upon the onset of pupation, live dorsal head epithelial tissue was dissected from *O. taurus* individuals in molecular-grade 1X PBS (phosphate-buffered saline) in sterilized 3-well glass dissection plates as described in ref. [12]. After dissection, an estimated 50,000 cells (determined by counting DAPI stained cells in a hemocytometer) were immediately transferred to a 1.5 mL conical tube and processed for ATAC-sequencing [71] using the Omni-Seq protocol [72]. Sequencing libraries were generated by amplifying open chromatin fragments with PCR, having determined the optimal number of amplification cycles with qPCR, as described in ref [73]. Sample were sequenced on an Illumina NextSeq 550 instrument to a minimum depth of 21.8 million 50 bp paired-end reads per sample, including a median 34.8 million reads after trimming and filtering per sample (17.4 million proper pairs) with Trimmomatic v. 0.39 (parameters: leading:10 trailing:10 slidingwindow:4:15 minlen:25) [74].

Filtered, paired-end ATAC-seq reads were aligned to the *O. taurus* genome assembly described above with Bowtie v.2.2.5 [75]. Alignments were filtered for minimum mapping quality (MAPQ) score of 20 and PCR duplicates were removed with Picard Tools v.2.18.7 (broadinstitute.github.io/picard). After all filtering steps, a median of 14.2 million high quality alignments remained across all samples. ATAC-seq peaks (hereafter open chromatin regions: OCRs) were called with macs2 v.2.2.6 [76] using the parameters:—nomodel—keep-dup = auto —shift 100—extsize 200 -g 2.9e8 -f BAMPE -q 0.05. All OCR sets were merged into a consensus set of 83,925 OCRs with the bedtools v.2.30.0 *merge* command (d = 0) [77], and peak accessibility counts were calculated with the bedtools *multicov* command.

## ATAC-seq Data analysis

Most statistical analyses of the ATAC-seq count data were carried out in R v.4.2.1. First, OCRs with extremely low accessibility counts were removed (3 counts-per-million required in at least 5 biological replicates), leaving a final set of 68,038 OCRs for downstream analysis (S4 Data). Differential accessibility analyses between sample groups (i.e. high [major] and low [minor] nutrition males, females) was calculated in DESeq2 v.1.36.0 [78] wherein significantly differentially accessible OCRs were called as having false-discovery rate support < 10%. OCRs were putatively assigned to a gene if it was located within 25 Kb (up- or downstream) of the gene's translational start site. While the transcription start site would normally be a more appropriate genomic reference for assigning OCRs to nearby coding sequence, annotation of

these features is currently not sufficient for every gene in the *O. taurus* genome, so the translation start site was instead chosen as a shared reference point among all gene models.

To identify orthologous OCRs across beetle species, a whole genome alignment was carried out on the soft repeat-masked genome assemblies of the ten beetle species analyzed in this study using Cactus v.2.3 [79]. The HAL file generated from this whole-genome alignment along with OCR coordinates were input into HALPER [80] with the parameters: -max_frac 4 -min_len 30 -protect_dist 1 to identify orthologous OCRs of *O. taurus* dorsal head epithelia in the genomes of each of the other nine beetle species analyzed. Motif enrichment analyses of OCR sets of interest were carried out in *homer* v.4.11 [81] with the parameters: -50,50 -mset insects -fdr 10 on a background set of the remaining (non-selected) OCRs identified in this study. Of note, validated transcription factor binding motifs are not available for *O. taurus* and enrichment analyses were searched against a database of *Drosophila melanogaster*. As a result, the closest match of some enriched motifs is to transcription factors not present in the *O. taurus* genome (e.g. bicoid). Gene and repeat annotations, comparative genomic results, and ATAC-seq result files have been deposited on Dryad [82].

## Dryad Doi

http://dx.doi.org/10.5061/dryad.wdbrv15t8

## Supporting information

**S1 Data. Annotation of Orthogroup Membership Gene IDs.**
(TXT)

**S2 Data. Results of CAFE gene family evolution analyses.**
(XLSX)

**S3 Data. Results of HYPHY-BUSTED evidence of episodic diversifying selection analyses.**
(TXT)

**S4 Data. Results of ATAC-seq statistical analyses.**
(XLSX)

**S5 Data. List of top 10% *doublsex* binding motif hits to *Onthophagus taurus* genome.**
(TXT)

**S1 Fig. Comparison of lineage-specific and shared OCRs repetitive element content.** A) Percent of total OCRs containing at least one repetitive element. B) Number and class of repetitive element for the 100 most abundant repeats present in each OCR type.
(DOCX)

**S1 Text. List of commands describing major computational steps involved in this study.**
(TXT)

## Acknowledgments

We thank Erica M. Nadolski for beetle head and horn illustrations.

## Author Contributions

**Conceptualization:** Phillip L. Davidson, Armin P. Moczek.

**Data curation:** Phillip L. Davidson.

**Formal analysis:** Phillip L. Davidson.

**Funding acquisition:** Phillip L. Davidson, Armin P. Moczek.

**Investigation:** Phillip L. Davidson.

**Methodology:** Phillip L. Davidson.

**Project administration:** Armin P. Moczek.

**Resources:** Armin P. Moczek.

**Supervision:** Armin P. Moczek.

**Visualization:** Phillip L. Davidson.

**Writing – original draft:** Phillip L. Davidson, Armin P. Moczek.

**Writing – review & editing:** Phillip L. Davidson, Armin P. Moczek.

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
