## [Decision Letter · Decision Letter 0]

13 Nov 2023

Dear Dr Davidson,

Thank you very much for submitting your Research Article entitled 'Genome evolution and divergence in cis-regulatory architecture underlie condition-responsive development in horned dung beetles' to PLOS Genetics.

The manuscript was fully evaluated at the editorial level and by independent peer reviewers.   The reviewers and editors were all very positive about your manuscript. We agree that it is interesting and the analyses generally well performed. However, the reviewers have raised a number of issues (detailed below) that we would like to give you the opportunity to address. 

We therefore ask you to modify the manuscript according to the review recommendations. Your revisions should address the specific points made by each reviewer.

Yours sincerely,

Elizabeth J. Duncan

Guest Editor

PLOS Genetics

Kelly Dyer

Section Editor

PLOS Genetics

Reviewer's Responses to Questions

**Comments to the Authors:**

Reviewer #1: This is an interesting and well-presented manuscript investigating the genetic/genomic basis of a novel trait that is affected by both sex and nutrition relation polymorphism. The manuscript presents the genomes of three species of dung beetle, assembly and annotation data associated, extending to an evolutionary analysis of gene presence/absence. The authors then present ATAC-seq data to indicate chromatin availability, allowing them to investigate the regulatory landscape of these traits, and, through clever use of different species, dissect the influence of sex and diet on the growth of beetle horns.

In my opinion this is an excellent manuscript well deserving of publication in PLOG genetics. It is a careful analysis of the regulation of a novel trait- and thus adds a great deal to our understanding of how such traits evolve, and how they are environmentally regulated. This more comprehensive approach indicates the importance of cis-regulatory element evolution in beetle horns, and the authors final comments in the conclusions are a very important challenge to others in this field.

I have a few, probably, minor issues that I think the authors need to address before publication.

1) The first is the data on gene numbers in these species. The authors present three very good genomes of Dung beetles. The finding that one of these species has considerably more gene predicted than the others, or indeed other beetle genomes, is a surprise. If this is truly the case I hope the authors are following up on this work. In most of the rest of the analyses I don’t see how these extra genes are included and one might suggest that our tools for analysing gene loss gain etc in these situations are poor- so perhaps the importance of these genes is down played. It would be nice to see a paragraph or two discussing these genes.

It is also a serious possibility that these novel genes are a result of mis-annotation. Clearly annotations improve with high quality RNAseq data being added- and this is indicated in the methods as having happened. The reference however is to a paper with no RNAseq datasets in it- So I am unsure as to what datasets are being used- is it possible that imbalances in the RNAseq data sets or amount of RNAseq used in each genome annotation have led to the appearance of the extra genes? Can the authors please indicate where their RNAseq comes from, and perhaps provide a table so we can be clear what RNAseq data has gone into each annotation process.

2) The authors present BUSCO results as the key quality measure for their genomes. This is OK, though it would be nice to see others if available. BUSCO however comes in lots of different forms- the Authors don’t state what version of BUSCO they used, nor what dataset they tested against. This is important as it is relative easy to achieve 95% BUSCO scores if testing your genome against a eukaryote dataset, harder if it is a beetle one. Could the authors please provide this information?

3) In Figure 4 the authors indicate some key potential transcription factor binding sites in their data- one of these is labelled Bicoid. As I understand it the Bicoid gene is limited to certain groups in the Diptera (see https://doi.org/10.1002/bies.20285). What then is this binding site- is it Hox3 , or Orthodenticle? I assume this has been identified through a Drosophila data set which has brought up this motif and labelled it as it is. The authors should explain this, and if possible, indicate what protein they think might be binding this motif.

4) Given the authors have Hi-C data as well as ATAC-seq (and I know the Hi-C is from different tissues and stages so this may be pointless), but does the Hi C interaction map confirm or provide extra information about the chromatic accessibility data?

5) There are a lot of bioinformatic analyses in this manuscript- and the availability of data is good. What I can’t find is any availability of scripts. For a paper like this, I would hope to see a GitHub (or similar) archive where I could find the scripts to replicate these results. I think this is really important, given the outcomes of this paper, and the need to compare with similar novel or non novel traits. Providing code is the best way to ensure this research is reproducible and widely used.

Reviewer #2: Plz check the attached review.

Reviewer #3: This study examines the genetic basis of adaptation in dung beetles using a combination of comparative genomics and experimental genomic work on regulatory variation associated with evolution of nutritional plasticity of an ornament (horn size). To do so the authors generate three new chromosome level assemblies from different dung beetles (combining PacBio HiFi and Hi-C technology) and combine these genomes with already existing assemblies from other beetle species to examine the genome evolution (losses/gains of genes and selection on them) that might be associated with dung beetle specific adaptations. The main findings from this was the identification of 200-300 evolving orthogroups and around 100 losses from the dung beetle node. These rapidly involving orthogroups in dung beetle node include some interesting gene families such as the juvenile hormone expansion in O.taurus and the large variation in TOR copy number between O.taurus and O.sagittarius.

Indeed, the gene expansion of the TOR gene is very interesting both from the point of view of its known growth regulatory role but also from some recent research in seed beetles showing that the Y chromosome harbours copy number variation in TOR that contributes to differences in body size between different male Y haplotypes (Kaufmann et al 2023 MBE*).

The other part of the ms examines the chromatin landscape across different comparisons (horn size differences and sex differences) using ATAC seq to better understand the CREs involved in nutrition induced morphological differences. These data come from the epithelial cells of the head where males develop horns, an important consideration in the experimental design which makes interpretation and link to horn morphology more robust. The main finds here are the identification of both nutrition induced changes in CRE accessibility in males and differences between the sexes which led to the identification of some novel candidate genes that can be tested in a functional setting. Combining the OCR analysis with differential expression also seem to, at least partly, establish a correlation between the two for candidate genes identified (although not at the genome wide level) which to my knowledge is rarely done thus far. Enrichment of DA OCRs on the X chromosome for the large honred vs small horned and large horned vs female comparisons is interesting and also a bit surprising…

Overall I think this ms explores a lot of interesting questions surrounding the role of plasticity and its genetic basis and the design of the experiment and the interpretations seem largely sound to me (though see comments below).

Major issues:

- While the gene family expansion analysis across the different species are interesting I think at times too much is made about their involvement in nutritional plasticity. There are surely many differences between the compared species and so I do not think that, for example, the TOR finding can be attributable to species differences in evolutionary exaggeration and secondary loss of male polyphenism (see lines 233-236 and 254-257 for examples where this is claimed). Differences in body size between species could potentially also be linked to TOR copy number variation (Kaufmann et al 2023), just to give one alternative interpretation. I would therefore advise to be less speculative in how these findings are discussed and interpreted not to fall into adaptive storytelling. It is good this is brought up specifically on lines 245-248 but perhaps it should be emphasized throughout since also the species comparison between taurus and sagittarius is set up as identifying gene families involved in nutritional plasticity but there are surely many other species differences between them apart from nutritional plasticity..

- A FDR of 10% for calling differential accessibility is rather lenient (p617-619)… I think it would be useful to report at 5% level what you find as well and keep a table in the Supp mat for this. And using a FDR of 10% just further highlight the importance of interpreting these results with caution.

Minor:

- I think the title is too conclusive in claiming that the observed changes at both the genomic level and regulatory level underlie the condition dependent development. It should be revised to reflect the fact that these are after all correlative analyses

- line 274: do you mean the horns or other developmental phenotypes?

- I find Fig 2D quite difficult to interpret but dont have any suggestions on how it could be improved…

- Fig 3: is the GW significant line the FDR at 10% line? Using Bonferroni correction it should be -log10(0.05/68000) = 6.1, which would leave a significantly smaller amount of DA OCRs.

References:

1) https://academic.oup.com/mbe/article/40/8/msad167/7227908

**Have all data underlying the figures and results presented in the manuscript been provided?**

Reviewer #1: Yes

Reviewer #2: Yes

Reviewer #3: Yes

PLOS authors have the option to publish the peer review history of their article (what does this mean?). If published, this will include your full peer review and any attached files.

Reviewer #1: No

Reviewer #2: No

Reviewer #3: No

---

## [Decision Letter · Decision Letter 1]

1 Feb 2024

Dear Dr Davidson,

We are pleased to inform you that your manuscript entitled "Genome evolution and divergence in cis-regulatory architecture is associated with condition-responsive development in horned dung beetles" has been editorially accepted for publication in PLOS Genetics. Congratulations!

Yours sincerely,

Elizabeth J. Duncan

Guest Editor

PLOS Genetics

Kelly Dyer

Section Editor

PLOS Genetics

Comments from the reviewers (if applicable):

Thank you for your careful attention to the reviewers comments. I have been over the revised manuscript and your response to the reviewers and am very pleased to recommend your manuscript for publication.

Please note that one of the reviewers has a small comment regarding the scale used for the Manhattan plot (Fig. 3) as it is usual to present on a Log10 scale rather than natural Log. I think the significance of these OCRs is clear from the figure and the natural log is clearly indicated on the y-axis label, it may not be as intuitive for the reader as a log10 scale, but I am happy to leave it up to you whether you alter the axis on these figures.

Reviewer's Responses to Questions

**Comments to the Authors:**

Reviewer #1: The authors have done an excellent job at addressing my comments- I am happy to recommend publication.

Reviewer #2: Please check the attachment

Reviewer #3: I think the authors have done a good job in the revision of their ms but I have a comment regarding the statistical analysis related to fig 3 still and the comment by the authors on what is being represented.

It is the first time that I see a Manhattan plot where the authors use natural logs instead of 10 log scale and I do not understand why that is done... With log10 scale it is easy to understand that if p = 0.00001 -log10 will be 5 but when using natural log it is instead 11.5 and to me the use of natural log scale is misleading and difficult to interpret. E.g. how should we interpret a 5% or 10% FDR on adjusted values on a natural log scale? I think there is a high risk readers will glance at the figure and come away thinking it is on log10 scale and that the results are more robust than they are.

**Have all data underlying the figures and results presented in the manuscript been provided?**

Reviewer #1: Yes

Reviewer #2: Yes

Reviewer #3: Yes

PLOS authors have the option to publish the peer review history of their article (what does this mean?). If published, this will include your full peer review and any attached files.

Reviewer #1: **Yes: **Peter Dearden

Reviewer #2: No

Reviewer #3: No

**Data Deposition**

http://datadryad.org/submit?journalID=pgenetics&manu=PGENETICS-D-23-01079R1

**Press Queries**

---

## [Editor Report · Acceptance letter]

29 Feb 2024

PGENETICS-D-23-01079R1 

Genome evolution and divergence in cis-regulatory architecture is associated with condition-responsive development in horned dung beetles 

Dear Dr Davidson, 

We are pleased to inform you that your manuscript entitled "Genome evolution and divergence in cis-regulatory architecture is associated with condition-responsive development in horned dung beetles" has been formally accepted for publication in PLOS Genetics! Your manuscript is now with our production department and you will be notified of the publication date in due course.

With kind regards,

Anita Estes

PLOS Genetics

On behalf of:
